

# BiLSTM-enhanced legal text extraction model using fuzzy logic and metaphor recognition

Jia Chen

College of Law and Sociology, Qinghai Normal University, Xining, Qinghai, China

## ABSTRACT

The burgeoning field of natural language processing (NLP) has witnessed exponential growth, captivating researchers due to its diverse practical applications across industries. However, the intricate nature of legal texts poses unique challenges for conventional text extraction methods. To surmount these challenges, this article introduces a pioneering legal text extraction model rooted in fuzzy language processing and metaphor recognition, tailored for the domain of online environment governance. Central to this model is the utilization of a bidirectional long short-term memory (Bi-LSTM) network, adept at delineating illicit behaviors by establishing connections between legal provisions and judgments. Additionally, a self-attention module is integrated into the Bi-LSTM architecture, augmented by L2 regularization, to facilitate the efficient extraction of legal text information, thereby enabling the identification and classification of illegal content. This innovative approach effectively resolves the issue of legal text recognition. Experimental findings underscore the efficacy of the proposed method, achieving an impressive macro-F1 score of 0.8005, precision of 0.8047, and recall of 0.8014. Furthermore, the article delves into an analysis and discussion of the potential application prospects of the legal text extraction model, grounded in fuzzy language processing and metaphor recognition, within the realm of online environment governance.

## INTRODUCTION

With the exponential growth of the internet, an enormous volume of legal text data has been digitized and stored in electronic forms. These legal texts encompass a wide array of materials, such as regulations, case law, contracts, and lawyer documents, among others. Given their complexity, it is a daunting task for legal professionals to effectively analyze and comprehend these legal documents (*Liang et al., 2023*). For example, the large-scale text data makes manual processing both heavy and time-consuming, and the processing of legal documents requires more Rigorous professionalism and workflow. Therefore, the automatic processing and classification of legal texts by computer has become a research hotspot.

Legal text consists of various text types, including contracts, legal terms, judgments, regulations, and legal documents. These texts are often unstructured and may contain

Corresponding author
Jia Chen, chenjia041518@163.com

structural fields such as title, author, publication date, length, classification, *etc.*, as well as unstructured text components such as abstracts and body content, and in addition contain a large amount of natural language and legal terminology, which makes traditional manual processing methods difficult to use for large-scale legal text data. At the same time, due to the rapid development of today's social environment, a large number of text data itself will bring a lot of data processing problems, so it is necessary to put forward more advantageous processing methods for these legal text data. To address this problem, researchers have started to utilize natural language processing and machine learning techniques to automate the processing of legal text (*Bommarito, Katz & Detterman, 2021*). In recent years, deep learning technology has made significant progress in the field of natural language processing and can be employed to automate the processing and analysis of extensive legal text data. Deep learning methods use neural network models to extract features from vast amounts of legal text data and automate various tasks, such as text classification, information extraction, keyword extraction, and entity recognition. These tasks can help lawyers and legal professionals find relevant text information more quickly, identify and understand legal terms and regulations, and enhance their work efficiency and accuracy (*Thomas & Sangeetha, 2021*).

However, compared to general natural language processing, extracting legal text poses certain difficulties. Legal text has the characteristics of formality, complexity, and ambiguity (*Medvedeva, Vols & Wieling, 2020*). Firstly, legal text usually employs formal terminology, formats, and language, and these terms may not be common or may have different meanings in everyday language, necessitating explanation and translation. Additionally, legal text often involves complex legal concepts and procedures that require a profound understanding to accurately comprehend their meanings and purposes. Moreover, due to the complexity and formality of legal text, there may be ambiguity or fuzziness, necessitating explanation and clarification. Considering these characteristics, utilizing fuzzy language processing methods and metaphor recognition methods can help the model better extract and identify the meaning and purpose of legal text.

Fuzzy language refers to language that is not clear or specific enough in meaning (*Qahtan et al., 2023*). In legal text, fuzzy language can lead to the uncertainty of legal provisions, and the model needs to extract the corresponding meaning of legal text from the case law of legal practice during the recognition process of legal text. Specifically, the model is used to extract the relevant features of legal text data, identify and classify them, extract keywords and keyframes, and introduce self-attention module to strengthen the understanding of the meaning of legal text. Metaphor refers to language expressions that are not literal but represent a certain meaning through analogy or comparison. In legal text, metaphors are often used to explain legal concepts and rules, and understanding metaphors depends on the reader's background knowledge and contextual understanding ability. These two types of text are common in legal judgments and interpretations of legal text. The legal text extraction model needs to accurately identify fuzzy language and metaphors in the text when extracting text information (*Minaee et al., 2021*).

Existing research has effectively utilized a variety of deep learning models to tackle the challenges associated with legal text processing, employing architectures tailored to extract

features from complex and unstructured data. Convolutional neural networks (CNNs) have been applied for sentence-level legal text classification, leveraging their ability to capture local dependencies. Recurrent neural networks (RNNs), particularly long short-term memory (LSTM) networks, have been extensively used for sequence modeling, enabling the analysis of sequential dependencies within legal documents. Bidirectional LSTMs (Bi-LSTM) further enhance this by capturing both forward and backward contextual relationships, which are crucial for understanding the interplay between case descriptions and legal provisions. Attention mechanisms, including hierarchical attention networks (HAN) and self-attention, have been integrated to focus on key sections of text, improving classification accuracy and interpretability. Transformer-based models, such as BERT and its legal-specific variants like Legal BERT, have shown remarkable performance in tasks like named entity recognition and legal question answering due to their pre-trained contextual embeddings and fine-tuning capabilities.

However, despite these advancements, significant research gaps persist. Many existing approaches fail to establish a robust bidirectional relationship between case descriptions and legal judgments, leading to incomplete semantic comprehension. Additionally, few models address the ambiguity and metaphorical language often found in legal texts, which requires advanced natural language understanding capabilities. Furthermore, while transformer models excel in general natural language tasks, their application in legal text processing often overlooks domain-specific linguistic nuances and contextual dependencies.

Therefore, the purpose of this study is to explore the design of a legal text extraction model based on fuzzy language processing and metaphor recognition. We selected legal judgments in the field of web platform content and aimed to establish a connection between the practical content of law and the content of the web platform by designing a natural language processing-based legal text extraction method, thereby effectively identifying the risk of content on the web platform and helping to govern the network environment.

In this article, we incorporated self-attention into the bidirectional long short-term memory network (Bi-LSTM) model and conducted L2 regularization to extract legal text information. The Bi-LSTM is employed to connect the descriptions of illegal behavior in legal provisions and judgments bi-directionally. Self-Attention is an attention module that adds weighted calculations to the output sequence through an attention module added before the fully connected layer, allowing the model to better focus on important information in the text and improve the model's performance. We applied L2 regularization to the loss function to prevent overfitting. We conducted experiments using publicly available judgment texts as the dataset and analyzed the experimental results to demonstrate the feasibility and effectiveness of the method in practice.

In this article, we constructed a legal text extraction model based on fuzzy language processing and metaphor recognition and verified the effectiveness of this method through

experiments. The principal contributions of this article can be succinctly summarized as follows:

1) Introducing a novel approach for extracting legal text utilizing a Bi-LSTM and attention mechanism. The model establishes a correlation between legal text and case descriptions, enabling it to predict the relevant infringed laws with high accuracy.

2) Proposing a method to classify the risks of illicit content on network platforms based on the aforementioned model. The model's predictions and classifications can aid reviewers in expeditiously processing illicit information, thereby enhancing the efficacy of network environment governance.

3) Analyzing and discussing the potential applications of a legal text extraction model based on fuzzy language processing and metaphor recognition in network environment governance. It is posited that this approach can effectively anticipate the risks of illicit content on network platforms, reducing the review time and significantly elevating the level of network environment governance.

## RELATED WORKS

The rapid proliferation and progress of the internet has resulted in a surge of illicit information on online platforms, including but not limited to false advertisement, infringement of intellectual property, lewd pornography, and fraudulent schemes. These nefarious contents not only infringe upon the lawful rights and interests of users, but also have deleterious impacts on society as a whole. Therefore, it has become an imminent imperative to devise effective measures for regulating illicit information on online platforms.

Conventional approaches for monitoring illicit content primarily rely on manual reviews, which are inefficient, costly, and prone to human error. Prior to the advent of machine learning techniques, *Sobel (2014)* and his peers conducted studies on identifying unlawful behaviors using statistical and probabilistic methods (*Sobel, 2014*; *Ruger et al., 2004*; *Lauderdale & Clark, 2012*), which may result in the loss of crucial information such as case descriptions and partial prediction outcomes.

With the advancement of natural language processing technology, utilizing legal text information extraction methods to automatically regulate unlawful content on online platforms has emerged as a novel solution. Legal text information extraction is the process of extracting key content from legal documents. Firstly, the entire legal text should be carefully read to have an overall understanding of the structure and content of the text. Secondly, the definition and interpretation clauses in the legal text should be paid attention to. Extract important information such as the nature of illegal acts and the identity of offenders, and classify and summarize it, so as to realize the automatic supervision of illegal information on network platforms. This method boasts advantages such as high efficiency, accuracy, and reliability, which can effectively improve regulatory efficiency and reduce costs, while mitigating issues such as human error. Notably, *Sulea et al. (2017)* utilized machine learning methods to classify cases, while *Merchant & Pande (2018)* leveraged

machine learning techniques in the field of natural language processing to enhance the accuracy of legal text-related tasks. However, these works encountered issues such as losing semantic information in case descriptions. In response to these limitations, *Luo et al. (2017)* proposed legal judgment prediction methods based on deep learning, mainly utilizing neural network models like convolutional neural networks (CNN) or bidirectional recurrent neural networks (LSTM) to encode case descriptions. Nonetheless, these approaches failed to consider the relationship between legal text and case description information (*Zhong et al., 2018*; *Xu et al., 2004*). *Riya et al. (2021)* propose an automated legal model based on machine learning to provide the efficiency of legal support systems, but this is largely structured input and processing based on computer programs. *Taylor & Bengo (2021)* built semi-automatic tools and established a *corpus* for unstructured literature data, and used machine learning and other tools to provide the basis for case law classification. The data in this aspect is relatively one-sided, and most of it is the basic work in the early stage.

Subsequent deep learning approaches have been applied in the legal domain, with some studies focusing on legal judgment tasks. For instance, *Wang et al. (2019)* proposed a hierarchical text matching model for case descriptions and legal provisions to predict the suspect's crime. Additionally, some works have examined the conviction prediction accuracy of categories with few samples, such as *Hu et al. (2018)* introduced criminal charge features to enhance the conviction prediction accuracy of categories with few samples. Although these deep learning-based research works have achieved favorable outcomes in legal judgment prediction tasks, they have not taken into account the bidirectional relationship between legal text and case description information.

In the era of rapid development of online information platforms, traditional audit methods for online platforms typically require legal professionals to search for relevant laws and regulations for potential illegal information and make judgments based on professional knowledge. These methods are generally costly and fall short of meeting demand. Thus, *Fawei et al. (2018)* proposed an intelligent legal Q&A system that utilizes case information and legal reasoning to analyze and answer questions. *Long et al. (2019)* applied a reading comprehension model to assess and answer legal questions based on case information, legal regulations, and question requests. Meanwhile, *Xiao et al. (2017)* proposed using a CNN model to classify Chinese legal questions to help the general public acquire legal knowledge. Although these research results have provided consulting assistance to audit personnel to some extent, there is still a long way to go before achieving intelligent assisted audit.

These studies indicate that applying legal text content analysis methods to online platform content management holds great promise, but there are still issues that require attention. Therefore, this article proposes a legal text extraction method based on fuzzy language processing and metaphor recognition, which has delivered excellent results in legal text analysis and prediction classification accuracy.

## METHODS

The proposed model employs a Bi-LSTM neural network architecture as the foundation. Bi-LSTM networks are capable of processing input sequence data from both directions, capturing past and future context more effectively. This bidirectional processing enables the model to better understand the overall meaning and semantics of legal texts.

### Holistic process

On top of the Bi-LSTM layers, we integrate a multi-head self-attention mechanism. Self-attention allows the model to focus on the most relevant parts of the input texts by calculating attention weights between each pair of words. The multi-head extension creates multiple sets of attention weights from different subspaces, enriching the model's attentional capabilities. Specifically, we use eight parallel attention heads in our model.

The complete process of the proposed model is depicted in Fig. 1.

Figure 1 depicts the holistic process of our methodology. Our data comprises three interdependent elements, namely, the case description, applicable laws, and judgment results. The case description delineates the facts, evidence, and pertinent legal provisions, establishing the foundation for the application of laws and rendering judgments. The applicable laws construe and apply legal provisions and regulations intrinsic to the case, consequently determining the legal application and liability of the case. The judgment results embody the legal judgments made by the judge predicated on the facts, evidence, and applicable legal provisions. As can be seen from Fig. 1, the legal text is first preprocessed to obtain the corresponding feature vector, and then the Bi-LSTM network is introduced to obtain the feature vector of relevant information after model processing. After recognition and classification, the category of the legal text is finally output. To ensure a fair and equitable judgment, it is imperative to meticulously consider the case facts and evidence, apply legal provisions, and make sound judgments based on legal principles and precedents.

### Bi-LSTM

During the data preprocessing phase, we initially preprocess the textual content of the judgment and encode it into a vector $x_i$. Subsequently, we employ an attention mechanism to amalgamate the distinct legal elements' representations in the judgment result, culminating in the comprehensive vector representation of the judgment result.

$$v_i = \begin{cases} W_x x_i + b_x = 0 \\ -inf = 1 \end{cases} \tag{1}$$

$$a_i = \frac{\exp\left(v_i^{\mathrm{T}} u_x / \sqrt{d}\right)}{\sum_i \exp\left(v_i^{\mathrm{T}} u_x / \sqrt{d}\right)} \tag{2}$$

$$c = \sum_i a_i h_i. \tag{3}$$

Here, $W_x$, $b_x$ and $u_x$ correspond to parameter matrices. The vector representation of the input legal element, $x_i$, is obtained through encoding, where $d$ represents the dimension of

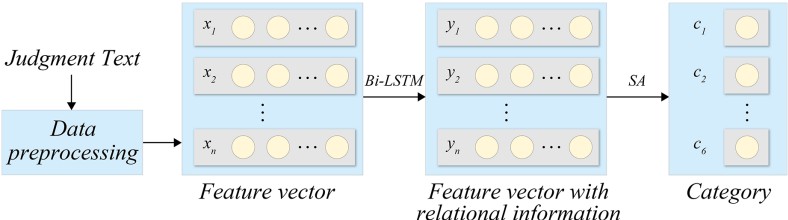

**Figure 1 Processing flow.**

the word vector in the input sequence. Additionally, $c$ denotes the comprehensive vector representation of all legal elements $h_i$.

Keeping this in mind, we have utilized the Bi-LSTM model to extract information from the judgment texts present in our dataset. Bi-LSTM, a type of bidirectional recurrent neural network, comprehensively analyzes the contextual information in the text by considering not only the context information before the current time step but also the context information after the current time step. In addition, the shared feature representation is realized through the shared bidirectional LSTM layer to improve the classification performance of the model. Specifically, Bi-LSTM comprises two LSTM networks, one processing the input sequence in a chronological order and the other processing it in a reverse order. For legal text data, information can be extracted from two different reading directions, one is from front to back, one is from back to front, and finally the output of the two networks is connected into a comprehensive output, which is used as the input of the next layer. Please refer to Fig. 2 for a diagrammatic representation of the Bi-LSTM model.

The initial layer in the diagram exemplifies a forward LSTM block, where the resulting output $h_t$ relies on both the current input $x_t$ and the prior output $h_{t-1}$. The subsequent layer in the diagram illustrates a backward LSTM block, in which the output $h_t$ is influenced by the current input $x_t$ and the succeeding output $h_{t+1}$. Hence, the present output $h_t$ is determined by $x_t$, $h_{t-1}$ and $h_{t+1}$. The computation formula for $h_t$ is expressed by Formula (4).

$$\vec{h}_t = LSTM(h_{t-1}, x_t) \tag{4}$$

$$\overleftarrow{h}_t = LSTM(h_{t+1}, x_t) \tag{5}$$

$$h_t = \alpha \vec{h}_t + \beta \overleftarrow{h}_t. \tag{6}$$

The symbols $\alpha$ and $\beta$ in the equation represent weight coefficients.

## Knowledge graph embedding

To enrich the semantic understanding of legal text data, we incorporated domain-specific knowledge using a knowledge graph embedding technique. The knowledge graph, constructed from legal documents and structured legal databases, captures the relationships between legal terminologies, case descriptions, and regulations. For this

**Peer**J Computer Science

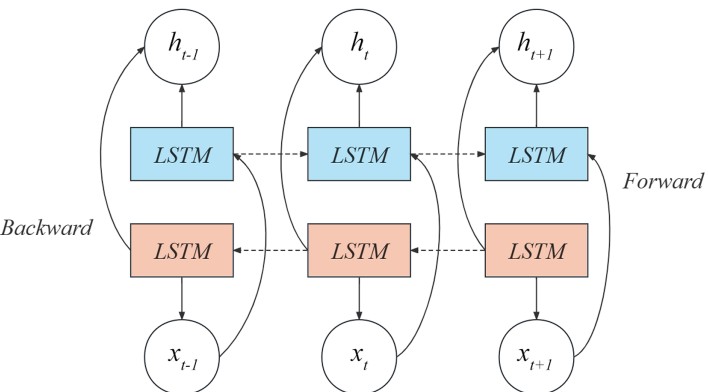

**Figure 2** Bi-LSTM network structure.

purpose, we utilized the TransE algorithm to embed the nodes and edges of the knowledge graph into a low-dimensional vector space.

In the TransE algorithm, the goal is to map entities (nodes) and relations (edges) in a knowledge graph into a low-dimensional space such that for a triple $(h, r, t)$, where $h$ is the head entity, $r$ is the relation, and $t$ is the tail entity, the following equation holds:

$$h + r \approx t \tag{7}$$

where $h, r, t$ are the vector representations of the head, relation, and tail entities in the low-dimensional vector space. The vector space is learned by minimizing the distance $d$ between $h + r$ and $t$.

To learn the embeddings, TransE uses a margin-based ranking loss function:

$$\mathcal{L}(h, r, t) = \sum_{(h'r,t') \in D} \max(0, \gamma + d(h + r, t) - d(h' + r, t')) \tag{8}$$

where $\gamma$ is a margin to separate positive and negative triples, $d$ is a distance measure (e.g., Euclidean distance) between the head + relation vector and the tail vector.

To align the knowledge graph embeddings with text embeddings, a common approach is to use cosine similarity. If $e_{text}$ is the embedding vector of the text and $e_{kg}$ is the embedding vector of the knowledge graph, the alignment loss can be represented as:

$$\mathcal{L}_{\text{align}} = 1 - \frac{e_{text} \cdot e_{kg}}{\| e_{text} \| \| e_{kg} \|} \tag{9}$$

where $e_{text} \cdot e_{kg}$ is the dot product of the text and knowledge graph embeddings, and the denominator normalizes the vectors to compute the cosine similarity. The goal is to minimize the alignment loss to ensure the consistency and complementarity between the two types of embeddings.

## Self-attention mechanism

After utilizing Bi-LSTM to extract contextual semantic features, we can deduce the legal article category to which the text vector pertains. In order to improve the recognition efficiency of key sentences, we add the self attention module after the Bi-LSTM feature

extraction and before the full connection layer. Self-attention is an extensively employed attention mechanism within natural language processing, serving to compute the interdependence amid elements within the input sequence. It facilitates calculating the extent of correlation between each element and other elements within the input sequence, thereby enabling a more comprehensive acquisition of the semantic information embedded within the input sequence.

The self-attention heads equip the model with an awareness of text structure and inter-dependencies between components of the legal text. This facilitates extracting pertinent features and making inferences about the semantic meaning more accurately. For example, the model can relate keywords in the case description to relevant statutes in the applicable laws section to predict appropriate judgment results.

During training, the datasets for case descriptions, applicable laws, and judgment results are fed into the model. The Bi-LSTM layers extract semantic features from the word sequences in each section. The self-attention head then identifies the most relevant interactions between features, which can establish global dependencies and expand the ability to extract text data. Finally, the representation vectors of the three parts are aggregated to make the final judgment prediction. Among them, the Bi-LSTM network model can better process the relevant data for the data set of case description, applicable law and judgment results. The advantage of self-attention block is that it can effectively combine elements at different positions in the input sequence, while considering their relative importance, so as to capture the interrelationship between elements in the sequence (*Pan et al., 2022*).

Taking into account the dataset's properties, we established the experimental configuration as follows: num_heads was set to 8, while query_dim, key_dim, and value_dim were set to 64. Additionally, output_dim was assigned a value of 128, and the dropout_prob was set at 0.1.

## EXPERIMENT

### Experimental setting

The implementation of the BiLSTM-enhanced legal text extraction model was conducted using a Linux-based operating system, specifically Ubuntu 20.04 LTS. The hardware setup included a high-performance Intel Core i7 processor, complemented by an NVIDIA RTX 3080 GPU with CUDA support to accelerate the training and inference processes. The system was equipped with 32 GB of RAM to handle the computational demands of large datasets and a 1 TB SSD for efficient data storage and quick access to model checkpoints. The software environment was built around Python 3.8, utilizing TensorFlow 2.x as the primary deep learning framework, with supporting libraries such as NumPy and Pandas for data manipulation, Scikit-learn for preprocessing and evaluation metrics, and Matplotlib for visualizations. GPU acceleration was facilitated through CUDA and cuDNN, while Jupyter Notebooks provided an interactive platform for experimentation and result analysis.

## Dataset

In the experiment, we utilized a dataset (doi: 10.5281/zenodo.11057826) procured from the internet, comprising of 487 judgments concerning the governance of online environments. The dataset was classified into three distinct categories of charges, each of which was further subcategorized based on the severity of the judgment results, totaling to six categories. For the purpose of simplicity in nomenclature and computation, we represented the three categories of charges as a, b, and c, while the less severe judgment results were denoted by 0, and the more severe ones were represented by 1.

## Data preprocessing

During the data preprocessing phase, a series of meticulous steps were implemented to prepare the legal text data for effective input into the Bi-LSTM-based extraction model. First, the raw legal text was normalized by removing extraneous spaces, special characters, and unnecessary formatting to ensure uniformity and cleanliness of the data. The text was then tokenized, splitting the continuous text into individual words or phrases to facilitate subsequent feature extraction. After tokenization, stop words (*e.g.*, common function words like "is" or "and") were removed to reduce noise and allow the model to focus on meaningful content.

Next, Word2Vec was employed to encode the tokenized text into numerical vectors. These word vectors preserved semantic relationships among words and helped capture contextual information. During this process, term frequency filtering was applied to remove overly frequent or rare terms, which could potentially distort the learning process. Subsequently, the data was organized into three core elements: case descriptions (facts and evidence), applicable laws (specific legal provisions), and judgment results (conclusions based on facts and laws). This structured categorization enabled the model to extract features from multiple dimensions.

To capture long-range dependencies and enhance semantic understanding, a self-attention mechanism was integrated prior to feature extraction. This mechanism calculated weighted relationships between each word in the input sequence, allowing the model to focus on significant information and generate a comprehensive vector representation. To further improve the model's generalization capability, L2 regularization was applied during feature preparation to mitigate overfitting. Additionally, all feature vectors were normalized to ensure uniform data distribution and stable feature representation.

The dataset was segregated into a training set and a test set, maintaining a 7:3 ratio, with 341 and 146 samples, respectively. Table 1 illustrates the distribution of samples across each category.

To address the issue of dataset size, we employed data augmentation techniques to enhance the dataset and improve the robustness of the model. Specifically, we implemented the following augmentation strategies. (1) Synonym replacement: Randomly selected words in the case descriptions were replaced with synonyms based on a predefined legal thesaurus, preserving the semantic meaning while diversifying the dataset. (2) Paraphrase generation tools were used to create alternate sentence structures for case

Table 1 The number of different categories of data.

| Category | Training | Testing | Total |
|----------|----------|---------|-------|
| a0 | 50 | 22 | 72 |
| a1 | 60 | 22 | 82 |
| b0 | 56 | 21 | 77 |
| b1 | 62 | 25 | 87 |
| c0 | 58 | 30 | 88 |
| c1 | 55 | 26 | 81 |

descriptions and applicable laws, ensuring variety while retaining legal and contextual accuracy. (3) Longer legal texts were divided into smaller segments, while shorter case descriptions were concatenated with related supplementary data, providing a wider variety of sentence lengths and structures. These techniques increased the dataset size from 487 samples to 1,461 samples, achieving a threefold expansion of the data. By diversifying the dataset, we aimed to improve the model's ability to generalize and handle variations in legal language, as shown in Table 2.

## Results

We utilized the t-SNE algorithm to diminish the dimensionality of text-encoded vectors in the dataset, in order to visually demonstrate the disparities present within. The resultant visualizations are depicted in Fig. 3. As shown in the figure, there are clear boundaries between the different categories, especially in the upper right and lower left corner, with relatively dense data between points, compared to relatively small data boundaries in the middle and scattered data between points.

The vector resulting from encoding the text and subjecting it to Self-Attention computation is predisposed towards key phrases. To avert overfitting of the text vector's classification results, we incorporated the L2 regularization method into the loss function. The resulting modified cross-entropy loss function is presented in Formula (10).

$$Loss = -\sum_{i=0}^{n} y_i log\left(\hat{y}_i\right) + \lambda \sum_{i=0}^{n} \| w_i \|_2^2 . \tag{10}$$

In the given formula, the symbol $\hat{y}_i$ denotes the anticipated classification of the degree of unlawfulness pertaining to the case description, $y_i$ signifies the actual value present in the dataset, $\lambda$ stands for the regularization coefficient, and $w_i$ corresponds to the weight parameter of the model. Upon conducting several iterations, we ascertained that establishing the regularization coefficient $\lambda$ as 0.01 facilitates the attainment of superior outcomes.

After preprocessing the data, we can obtain the feature vectors for legal texts. The training dataset comprises the feature vectors for case descriptions and their corresponding judgment categories. These vectors can be utilized to train the Bi-LSTM model. The

**Table 2 Has been updated to reflect the augmented dataset distribution.**

| Charge category | Less severe judgment (0) | More severe judgment (1) | Total |
| --- | --- | --- | --- |
| a | 250 | 260 | 510 |
| b | 200 | 215 | 415 |
| c | 255 | 281 | 536 |
| Total | 705 | 756 | 1,461 |

variation of the loss function during the training process of the Bi-LSTM model with increasing training epochs is demonstrated in Fig. 4.

From the depicted graph, it can be observed that the loss function value diminishes with the increase of training epochs during the model's training phase. Initially, owing to the random parameter initialization, the loss function value is relatively substantial, approximately 3.5. Nevertheless, the rate of decline of the loss function is swift, and it stabilizes after approximately 60 epochs. Subsequently, it gradually diminishes and attains the minimum value at 152 epochs. At this juncture, the model exhibits commendable proficiency in category prediction.

Subsequent to the training, the model's efficiency is evaluated using the test set data, and the corresponding confusion matrix is presented in Fig. 5.

The figure illustrates that the case classification problem tackled in this article pertains to a multi-classification predicament, with a confusion matrix size of 6 * 6. Notably, the article utilizes macro-F1 as the evaluation metric to assess the model's clustering efficacy. Macro-F1 is derived from the F1-score of binary classification. The approach entails dividing the multi-classification confusion matrix into multiple binary classification matrices, based on the binary classification method's "yes" and "no" criteria, computing several F1-scores, and subsequently averaging them to obtain the Macro-F1 value (*Sun et al., 2020*). The formula for macro-F1 calculation, as shown in Formula (11), is utilized.

$$\text{Precision} = \frac{TP}{TP + FP} \tag{11}$$

$$Recall = \frac{TP}{TP + FN} \tag{12}$$

$$F_1 - score = \frac{2 \times Recall \times Precision}{Recall + Precision} \tag{13}$$

$$Macro - F_1 = mean(F_{1\_i}). \tag{14}$$

In this context, *mean* refers to computing the arithmetic mean, while $F_{1\_i}$ denotes the F1-score of the *i*-th binary classification matrix. Utilizing the confusion matrix illustrated in Fig. 5, it can be converted into several binary classification matrices.

Precision, recall, and F1-score are computed for each of the binary classification matrices, and subsequently, the evaluation values of the multiple binary classification matrices are averaged to derive the evaluation standard for the multiple problems.

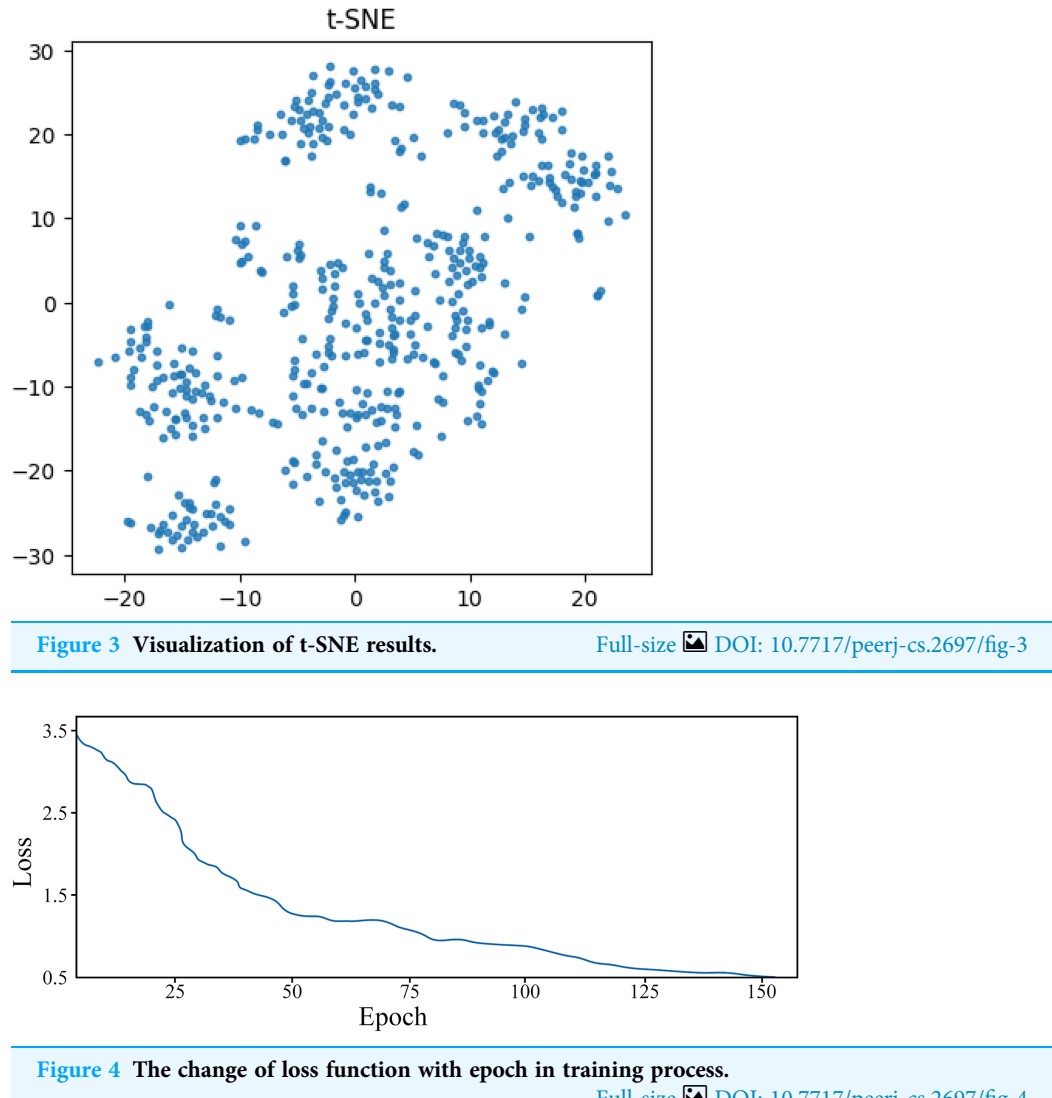

**Figure 3 Visualization of t-SNE results.**     

**Figure 4 The change of loss function with epoch in training process.**

Table 3 exhibits the classification efficacy of the model on the test set, predicated on diverse binary confusion matrices. The test set encompasses 146 samples, and the evaluation metrics for distinct categories are delineated in the table. The outcomes evince that the model's performance is optimal in categorizing c1, with an accuracy of 0.9091. This is owing to the fact that c1 corresponds to the most severe law and encompasses the most potent linguistic depiction in the judgment text.

To compare the performance of different methods, we also deployed other information extraction and classification methods and conducted ablation experiments specifically for the proposed model in this article. Table 3 and Fig. 6 show their respective evaluation results.

VGG (Visual Geometry Group) (*Koonce & Koonce, 2021*): Suitable for basic visual tasks such as image classification, using a convolution kernel of small size (3 × 3) and deep network structure.
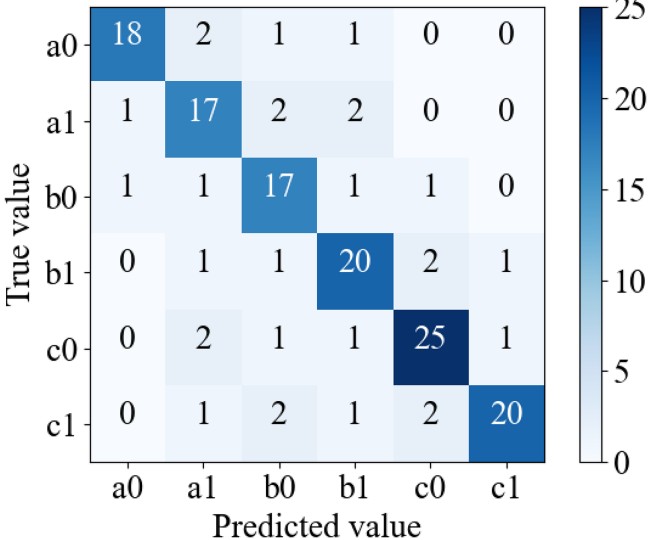

**Figure 5 Confusion matrix.**

**Table 3 Evaluation of the model on the test set.**

| Category | Precision | Recall | F1-score |
|---|---|---|---|
| a0 | 0.9000 | 0.8182 | 0.8571 |
| a1 | 0.7083 | 0.7727 | 0.7391 |
| b0 | 0.7083 | 0.8095 | 0.7556 |
| b1 | 0.7692 | 0.8000 | 0.7843 |
| c0 | 0.8333 | 0.8333 | 0.8333 |
| c1 | 0.9091 | 0.7692 | 0.8333 |

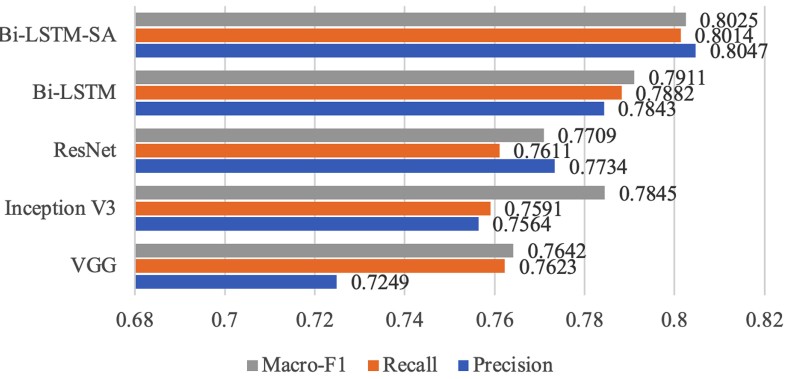

**Figure 6 Results of different models.**

Inception V3 (*Wang et al., 2019*): Convolution operations using multiple convolution nuclei of different sizes help capture features at different scales.

ResNet (Residual Network) (*Targ, Almeida & Lyman, 2016*): Uses residual blocks to allow information to flow directly to deeper networks by skipping connections, helping to train very deep networks.

BiLSTM (Bidirectional Long Short-Term Memory) (*Siami-Namini, Tavakoli & Namin, 2019*): A variant of recurrent neural networks (RNN) that helps capture contextual information in timing data by introducing hidden layers in both forward and backward directions.

Table 4 and Fig. 6 present the results of several methods evaluated by Macro-F1 as a performance metric. The outcomes signify that the Bi-LSTM model with Self-Attention, propounded in this article, surpasses the performance of other methodologies in predicting the classification of illicit content on online platforms. Compared to the VGG, Inception V3, and ResNet models, the Bi-LSTM-SA method proposed in this article discloses an improvement of 11.0%, 6.4%, and 4.0% in precision, 5.0%, 5.5%, and 5.2% in recall, and 4.8%, 2.0%, and 3.8% in macro-F1, correspondingly. In the ablation experiments, we employed the raw Bi-LSTM model for evaluation to demonstrate the effectiveness of the SA module. The experimental results showed that the evaluation performance of Bi-LSTM-SA was superior to that of Bi-LSTM. In summary, the experimental results confirm the effectiveness and robustness of the proposed model in this task. The Bi-LSTM-SA model proposed in this article can effectively learn the global dependency relationship of the text by adding self-attention mechanism on the basis of Bi-LSTM, focusing on the semantic information that is more important for the classification task, and thus achieving significant improvement in illegal content classification task compared with VGG, Inception V3, ResNet and the original Bi-LSTM model. This fully demonstrates the effect of self-attention mechanism on sequence modeling, as well as the innovation and effectiveness of the proposed model. This research provides valuable experience and further optimization directions for using deep learning technology to detect and filter illegal content on the Internet.

To further evaluate the interpretability of the proposed Bi-LSTM model with self-attention, we generated attention heatmaps for sample inputs, as shown in Fig. 7. These visualizations reveal how the model dynamically allocates attention weights to critical components of legal text, such as key terms, legal articles, and contextual phrases. For instance, in Heatmap 1, the model assigns high attention to terms like "violated" and "Article," highlighting its focus on capturing the explicit relationship between case descriptions and applicable laws. In contrast, Heatmap 2 shifts its emphasis to "defendant" and "21," demonstrating its ability to adapt attention based on the contextual significance of the legal entity and specific law sections. Heatmap 3 presents a more balanced distribution of attention across "Article" and "Penal Code," reflecting the model's capacity to process broader contexts for holistic legal analysis. Finally, Heatmap 4 shows concentrated attention on "21" and "Penal Code," underscoring the model's precision in referencing specific legal statutes. These results underscore the effectiveness of the self-

**Table 4 Comparison of multiple different models.**

| Method | Precision | Recall | Macro-F1 |
|---|---|---|---|
| VGG | 0.7249 | 0.7623 | 0.7642 |
| Inception V3 | 0.7564 | 0.7591 | 0.7845 |
| ResNet | 0.7734 | 0.7611 | 0.7709 |
| Bi-LSTM | 0.7843 | 0.7882 | 0.7911 |
| Bi-LSTM-SA | 0.8047 | 0.8014 | 0.8025 |

attention mechanism in capturing nuanced semantic relationships in legal text, enhancing both classification accuracy and interpretability.

## DISCUSSION

In the current epoch of rapid advancements in the field of artificial intelligence, machine learning and deep learning, in particular, have been widely employed by researchers in diverse fields. The language processing model examined in this article represents a promising avenue for legal text extraction. Initially, as language models continue to mature and evolve, researchers and engineers in related domains have extensively explored and experimented, proposing numerous exceptional language models and associated algorithms. These new models and algorithms have demonstrated greater precision and efficiency in various natural language processing tasks, significantly enhancing the efficacy of natural language processing technology in practical applications (*Shankar & Parsana, 2022*). Furthermore, with the consistent upgrading of hardware devices and the amelioration of computing power, modern language models' training and inference speed have been substantially enhanced, further promoting the widespread application and popularization of language models in practical settings. Therefore, it can be stated that the efficiency of related work has been substantially improved with the maturity of language models (*Madsen, Reddy & Chandar, 2022*). Secondly, in the area of legal text processing, conventional language models may confront some challenges when processing legal text due to its complexity and professionalism. Thus, it is essential to further explore and optimize language models in legal text processing to better adapt to legal text's unique content and language style. In this experiment, we used Bi-LSTM combined with self-attention for legal text information extraction and employed the L2 regularization approach to prevent overfitting of the training results. The experimental results revealed that this method's macro-F1 value can reach 0.8005, indicating that this method is more accurate than other approaches and is an effective method. The trained model can be used for the classification of illegal information in network platform content, assisting the platform in enhancing its ability to govern the network environment. However, the application of self-attention mechanism in natural language processing may cause memory explosion due to processing too long text, which is a problem that needs to be challenged.

Research and application in the field of legal text processing hold vast potential and prospects. With the internet and digital technology's development, the volume and diversity of legal texts continue to increase. Language models and natural language

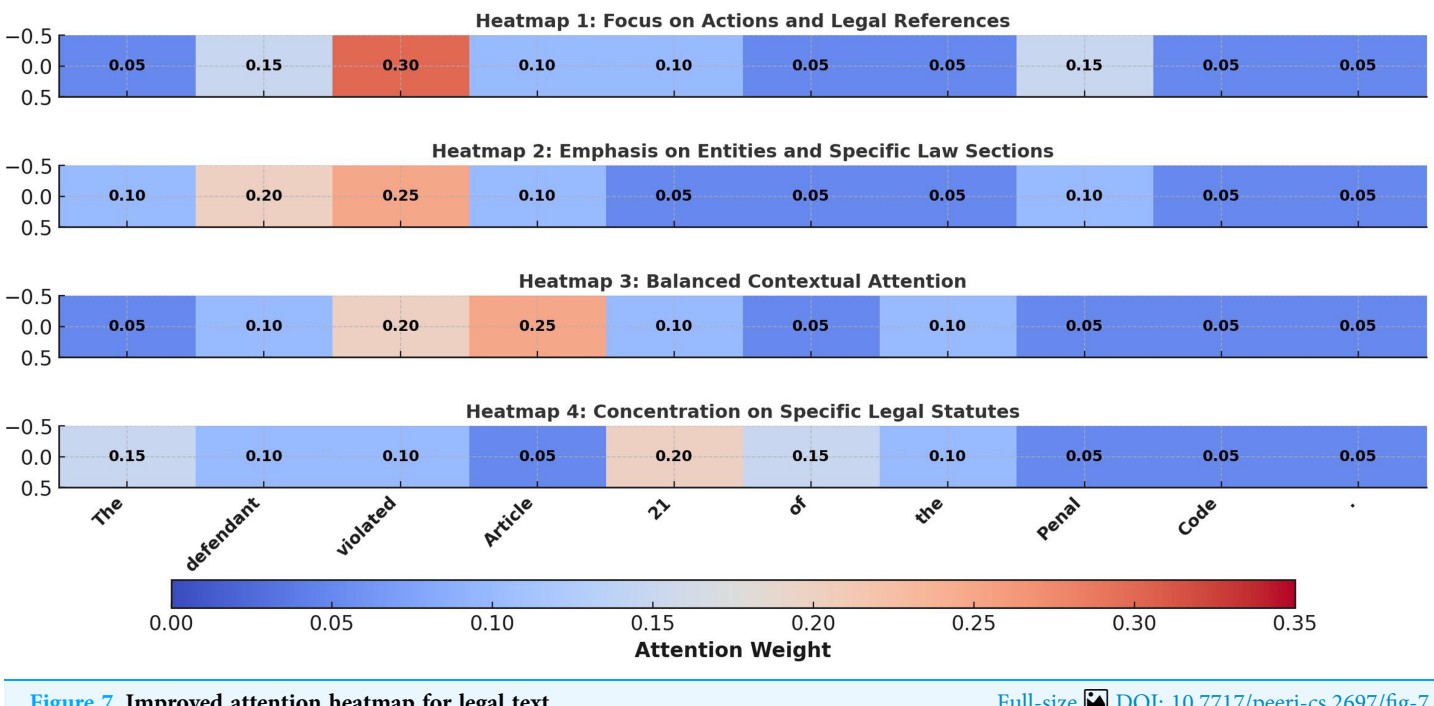

**Figure 7 Improved attention heatmap for legal text.**

processing technology can significantly enhance legal text processing and analysis efficiency and accuracy, reduce labor costs, and improve work efficiency. Moreover, legal text, as a crucial foundation for network governance and legal support, is of significant importance for network governance. In this regard, language models and natural language processing technology can assist network governance institutions in better processing and analyzing legal text, thereby better maintaining network order and protecting user rights. For instance, in addressing issues such as online speech, online infringement, and online fraud, language models and natural language processing technology can be deployed to analyze and explore relevant legal text, thereby better adjudicating and handling associated events (*Casanovas, de Koker & Hashmi, 2022*). Furthermore, language models and natural language processing technology can be utilized to develop intelligent network governance tools and systems to improve governance efficiency and accuracy. The model can improve the functionality of search engines for legal texts, particularly for queries involving vague or metaphorical expressions. Its ability to capture contextual meaning and resolve ambiguities ensures that the search results are more aligned with user intent, even in complex queries.

In the future, research in the field of legal text processing and its application in network governance will be of great significance (*Ihsan & Bintarsari, 2021*). While the performance of the model is partially constrained by the size and diversity of the dataset used for training and evaluation. Legal texts are often domain-specific and vary significantly across regions, languages, and legal systems. The current dataset, while representative, may not encompass the full variety of legal language, limiting the generalizability of the model to

other legal domains. Expanding the dataset to include a broader range of legal texts, such as case law, statutes, and legal contracts, would enhance the robustness of the model. A detailed error analysis of the model's performance indicates that certain misclassifications occur in cases involving highly ambiguous terms or overlapping semantic contexts. These errors highlight the need for more advanced contextual disambiguation techniques, such as incorporating knowledge graphs or external domain-specific ontologies.

## CONCLUSION

This study primarily investigates a legal text extraction model based on fuzzy language processing and metaphor recognition and explores the potential applications of this approach for online environment governance. By incorporating Bi-LSTM and a self-attention mechanism, the proposed model achieved a macro-F1 score of 0.8005, outperforming other models in precision and recall metrics. Additionally, data augmentation techniques were employed to mitigate the limitations posed by the small dataset size, enhancing the model's robustness.

However, this article has certain limitations that warrant further exploration. One challenge lies in scaling the model to larger and more diverse datasets. While the current model performs well on the selected dataset, its efficiency and performance may degrade when handling datasets with significantly more samples or diverse legal domains. For larger datasets, memory inefficiencies caused by the self-attention mechanism need to be addressed. Future work could explore memory-efficient variants of the attention mechanism, such as sparse attention or linearized attention models, to enable scalability without sacrificing performance. Additionally, the model faces challenges in accurately recognizing fuzzy language and metaphorical expressions, particularly in nuanced legal contexts. For instance, failure cases were observed when the context of a metaphor depended on domain-specific legal knowledge not directly captured in the dataset. These errors suggest that the model may benefit from integrating external legal knowledge sources, such as ontologies or pre-trained legal embeddings, to enrich its understanding of domain-specific language.

## ACKNOWLEDGEMENTS

The author would like to thank the anonymous reviewers for their valuable comments on this article.

### Funding
The authors received no funding for this work.

### Competing Interests
The authors declare that they have no competing interests.

## Author Contributions

• Jia Chen conceived and designed the experiments, performed the experiments, analyzed the data, performed the computation work, prepared figures and/or tables, authored or reviewed drafts of the article, and approved the final draft.

## Data Availability

The data is available at Zenodo: None. (2024). CAIL Judicial Summary—Dataset [Data set]. Zenodo. https://doi.org/10.5281/zenodo.11057826.

## Supplemental Information

Supplemental information for this article can be found online at http://dx.doi.org/10.7717/peerj-cs.2697#supplemental-information.

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
