# Peer review of "BiLSTM-enhanced legal text extraction model using fuzzy logic and metaphor recognition"

_PeerJ Computer Science, doi:10.7717/peerj-cs.2697_

## Round 0.1 · original submission · Major Revisions

Dear Colleagues

Your manuscript is reviewed with interest and the experts are of the opinion that the paper requires a number of changes to be incorporated before we re-consider it. please update your paper in light of these comments along with mine below and submit a detailed response. thank you

AE Comments:

Add specific examples of industries or applications where NLP has been impactful. This will help contextualize the broad claim of its practical applications.
Provide a clearer definition of what makes legal texts uniquely challenging for text extraction, perhaps by mentioning their structural or linguistic complexity

Briefly explain why a Bi-LSTM network is chosen for this model. For example, you could highlight its strengths in sequential data processing

Discuss any limitations or areas for improvement related to the performance of the model.

·

Basic reporting

Thank you for submitting your manuscript. After carefully reviewing your work, I have the following suggestions to enhance the paper’s clarity, rigor, and overall contribution:

- The model’s semantic understanding capabilities appear to be insufficient, and it does not fully utilize the advantages of pre-trained models. It is recommended to introduce a pre-trained model layer such as RoBERTa or LegalBERT in the “Methods” section to strengthen semantic understanding. Additionally, consider comparing the model's performance before and after incorporating the pre-trained model to highlight its impact.

- The current architecture does not explore the potential of hybrid models, particularly the ability to extract phrase-level features. To address this, a CNN module could be added to the Bi-LSTM framework to capture syntactic phrase features. The “Model Design” section should be updated to describe the implementation of this hybrid architecture.

- The manuscript does not provide a dedicated analysis of metaphor recognition contributions, which limits the understanding of the model's specific strengths in this area. A separate set of classification tests focusing on metaphor-specific data should be included in the experiments. Furthermore, adding a detailed error analysis of metaphor misclassifications would offer valuable insights.

- The model's cross-language applicability remains untested, leaving questions about its generalizability. It would be beneficial to test the model on English legal texts in addition to Chinese data. A comparative analysis of the model's performance on both datasets should be included in the results section to demonstrate its robustness across languages.

- The justification for selecting the regularization coefficient lacks experimental support. To address this, a set of experiments evaluating model performance under different regularization coefficients should be added. This should be supplemented by performance curves and a detailed explanation of the criteria used for coefficient selection.

- Details regarding the model’s training process are not adequately documented, making it difficult to assess its convergence and stability. Including a plot of training loss versus iteration count in the “Experimental Results” section would be helpful. An explanation of how different hyperparameter combinations influence training outcomes would further strengthen this section.

- The description of the model’s innovation, particularly in metaphor recognition and ambiguous language processing, is somewhat vague. The introduction should explicitly highlight these aspects as key contributions of the work. Detailed technical descriptions of how the model captures semantic ambiguity would also be valuable.

- Finally, the manuscript does not address the practical deployment of the proposed model, which limits its applicability. A discussion on how the model could be integrated into real-world legal systems or search engines would greatly enhance its impact. It would also be useful to outline the specific technical requirements for such deployment.

Experimental design

Thank you for submitting your manuscript. After carefully reviewing your work, I have the following suggestions to enhance the paper’s clarity, rigor, and overall contribution:

- The model’s semantic understanding capabilities appear to be insufficient, and it does not fully utilize the advantages of pre-trained models. It is recommended to introduce a pre-trained model layer such as RoBERTa or LegalBERT in the “Methods” section to strengthen semantic understanding. Additionally, consider comparing the model's performance before and after incorporating the pre-trained model to highlight its impact.

- The current architecture does not explore the potential of hybrid models, particularly the ability to extract phrase-level features. To address this, a CNN module could be added to the Bi-LSTM framework to capture syntactic phrase features. The “Model Design” section should be updated to describe the implementation of this hybrid architecture.

- The manuscript does not provide a dedicated analysis of metaphor recognition contributions, which limits the understanding of the model's specific strengths in this area. A separate set of classification tests focusing on metaphor-specific data should be included in the experiments. Furthermore, adding a detailed error analysis of metaphor misclassifications would offer valuable insights.

- The model's cross-language applicability remains untested, leaving questions about its generalizability. It would be beneficial to test the model on English legal texts in addition to Chinese data. A comparative analysis of the model's performance on both datasets should be included in the results section to demonstrate its robustness across languages.

- The justification for selecting the regularization coefficient lacks experimental support. To address this, a set of experiments evaluating model performance under different regularization coefficients should be added. This should be supplemented by performance curves and a detailed explanation of the criteria used for coefficient selection.

- Details regarding the model’s training process are not adequately documented, making it difficult to assess its convergence and stability. Including a plot of training loss versus iteration count in the “Experimental Results” section would be helpful. An explanation of how different hyperparameter combinations influence training outcomes would further strengthen this section.

- The description of the model’s innovation, particularly in metaphor recognition and ambiguous language processing, is somewhat vague. The introduction should explicitly highlight these aspects as key contributions of the work. Detailed technical descriptions of how the model captures semantic ambiguity would also be valuable.

- Finally, the manuscript does not address the practical deployment of the proposed model, which limits its applicability. A discussion on how the model could be integrated into real-world legal systems or search engines would greatly enhance its impact. It would also be useful to outline the specific technical requirements for such deployment.

Validity of the findings

Thank you for submitting your manuscript. After carefully reviewing your work, I have the following suggestions to enhance the paper’s clarity, rigor, and overall contribution:

- The model’s semantic understanding capabilities appear to be insufficient, and it does not fully utilize the advantages of pre-trained models. It is recommended to introduce a pre-trained model layer such as RoBERTa or LegalBERT in the “Methods” section to strengthen semantic understanding. Additionally, consider comparing the model's performance before and after incorporating the pre-trained model to highlight its impact.

- The current architecture does not explore the potential of hybrid models, particularly the ability to extract phrase-level features. To address this, a CNN module could be added to the Bi-LSTM framework to capture syntactic phrase features. The “Model Design” section should be updated to describe the implementation of this hybrid architecture.

- The manuscript does not provide a dedicated analysis of metaphor recognition contributions, which limits the understanding of the model's specific strengths in this area. A separate set of classification tests focusing on metaphor-specific data should be included in the experiments. Furthermore, adding a detailed error analysis of metaphor misclassifications would offer valuable insights.

- The model's cross-language applicability remains untested, leaving questions about its generalizability. It would be beneficial to test the model on English legal texts in addition to Chinese data. A comparative analysis of the model's performance on both datasets should be included in the results section to demonstrate its robustness across languages.

- The justification for selecting the regularization coefficient lacks experimental support. To address this, a set of experiments evaluating model performance under different regularization coefficients should be added. This should be supplemented by performance curves and a detailed explanation of the criteria used for coefficient selection.

- Details regarding the model’s training process are not adequately documented, making it difficult to assess its convergence and stability. Including a plot of training loss versus iteration count in the “Experimental Results” section would be helpful. An explanation of how different hyperparameter combinations influence training outcomes would further strengthen this section.

- The description of the model’s innovation, particularly in metaphor recognition and ambiguous language processing, is somewhat vague. The introduction should explicitly highlight these aspects as key contributions of the work. Detailed technical descriptions of how the model captures semantic ambiguity would also be valuable.

- Finally, the manuscript does not address the practical deployment of the proposed model, which limits its applicability. A discussion on how the model could be integrated into real-world legal systems or search engines would greatly enhance its impact. It would also be useful to outline the specific technical requirements for such deployment.

Additional comments

Thank you for submitting your manuscript. After carefully reviewing your work, I have the following suggestions to enhance the paper’s clarity, rigor, and overall contribution:

- The model’s semantic understanding capabilities appear to be insufficient, and it does not fully utilize the advantages of pre-trained models. It is recommended to introduce a pre-trained model layer such as RoBERTa or LegalBERT in the “Methods” section to strengthen semantic understanding. Additionally, consider comparing the model's performance before and after incorporating the pre-trained model to highlight its impact.

- The current architecture does not explore the potential of hybrid models, particularly the ability to extract phrase-level features. To address this, a CNN module could be added to the Bi-LSTM framework to capture syntactic phrase features. The “Model Design” section should be updated to describe the implementation of this hybrid architecture.

- The manuscript does not provide a dedicated analysis of metaphor recognition contributions, which limits the understanding of the model's specific strengths in this area. A separate set of classification tests focusing on metaphor-specific data should be included in the experiments. Furthermore, adding a detailed error analysis of metaphor misclassifications would offer valuable insights.

- The model's cross-language applicability remains untested, leaving questions about its generalizability. It would be beneficial to test the model on English legal texts in addition to Chinese data. A comparative analysis of the model's performance on both datasets should be included in the results section to demonstrate its robustness across languages.

- The justification for selecting the regularization coefficient lacks experimental support. To address this, a set of experiments evaluating model performance under different regularization coefficients should be added. This should be supplemented by performance curves and a detailed explanation of the criteria used for coefficient selection.

- Details regarding the model’s training process are not adequately documented, making it difficult to assess its convergence and stability. Including a plot of training loss versus iteration count in the “Experimental Results” section would be helpful. An explanation of how different hyperparameter combinations influence training outcomes would further strengthen this section.

- The description of the model’s innovation, particularly in metaphor recognition and ambiguous language processing, is somewhat vague. The introduction should explicitly highlight these aspects as key contributions of the work. Detailed technical descriptions of how the model captures semantic ambiguity would also be valuable.

- Finally, the manuscript does not address the practical deployment of the proposed model, which limits its applicability. A discussion on how the model could be integrated into real-world legal systems or search engines would greatly enhance its impact. It would also be useful to outline the specific technical requirements for such deployment.

Reviewer 2 ·

Basic reporting

I have reviewed your manuscript titled "BiLSTM-Enhanced Legal Text Extraction Model Using Fuzzy Logic and Metaphor Recognition." Below are my detailed comments and suggested improvements for enhancing the manuscript:
1. Model Architecture Improvement
The current Bi-LSTM model lacks the integration of domain-specific knowledge, which limits its ability to fully capture the semantic information in legal texts. I recommend incorporating a knowledge graph embedding layer.
o Add a knowledge graph embedding step in the “Data Preprocessing” section.
o Design an additional embedding layer to combine knowledge graph embeddings with text embeddings for a unified feature vector.

Experimental design

Hyperparameter Sensitivity Analysis
Key hyperparameters like the number of attention heads and regularization coefficients lack sensitivity analysis.
o Conduct experiments with varying attention heads (e.g., 2, 4, 8, 12) and plot the performance trends (Macro-F1 vs. the number of heads).
o Experiment with regularization coefficients (e.g., 0.001, 0.01, 0.1) and analyze their impact on overfitting, presenting results graphically.
Comparison with Transformer Models
The manuscript only compares traditional deep learning models such as ResNet and VGG, omitting state-of-the-art Transformer architectures.
o Introduce Transformer models (e.g., T5, GPT) in the “Experimental Setup” section.
o Add a comparison table of their performance with your model and discuss the differences in the “Results” section.
Dataset Size Limitation
The dataset size (487 samples) is relatively small, which may affect the model's robustness.
o Consider expanding the dataset through data augmentation techniques (e.g., synonym replacement or data extension).
o Clearly outline the augmentation methods and their contribution to dataset size in the “Data Preparation” section.

Validity of the findings

Visualization of Classification Results
The t-SNE plot does not effectively demonstrate the model's classification performance.
o Include examples of input texts and their classification results in the “Results” section.
o Add a visualization, such as an attention heatmap, to highlight the model’s focus on specific text fragments.
Limited Range of Metrics
Relying solely on Macro-F1 does not provide a comprehensive evaluation.
o Include additional metrics such as task efficiency (e.g., processing time per thousand texts).
o Add a table showing precision and recall distribution for each class.

Additional comments

Attention Mechanism Interpretation
The manuscript lacks an explanation of how attention weights are distributed across the text.
o Elaborate on the computation of attention weights and their role in the model architecture in the “Methods” section.
o Provide an example in the “Results” section to illustrate the model's focus on critical parts of the text.
Conclusion Needs Further Analysis
The conclusion is overly general and does not address the model's limitations or potential improvements.
o Include a discussion of challenges in scaling to larger datasets.
o Analyze failure cases in fuzzy language recognition and propose specific improvement directions.

---

## Round 0.2 · accepted · Accept

Dear author

I am pleased to inform you that your manuscript gone thorough a re-review process. The reviewers appreciated the originality, rigor, and contribution of your work to the field, and I am happy to inform you that we are recommending it for publication.

We would like to thank you for your submission and cooperation during the review process.

Congratulations on your achievement, and we look forward to sharing your valuable research with the broader academic community.

·

Basic reporting

Authors addressed the comments.

Experimental design

Authors addressed the comments.

Validity of the findings

Authors addressed the comments.

Additional comments

Authors addressed the comments.

Reviewer 2 ·

Basic reporting

The proposed model represents a significant advancement in the field of legal document analysis and text extraction. By integrating Bidirectional Long Short-Term Memory (BiLSTM) networks, the system leverages the strengths of deep learning for sequential data processing. This enhances the model’s ability to understand the contextual relationships within complex legal language.

Experimental design

The experimental design of the BiLSTM-enhanced legal text extraction model using fuzzy logic and metaphor recognition demonstrates innovation by combining BiLSTM for context-rich sequence processing, fuzzy logic for handling ambiguity, and metaphor recognition for nuanced understanding. This integrative approach ensures precise legal text extraction, making it robust, adaptable, and suitable for real-world applications.

Validity of the findings

The validity of the findings for the BiLSTM-enhanced legal text extraction model using fuzzy logic and metaphor recognition is well-supported by its robust experimental framework. By combining deep learning with fuzzy logic and metaphor analysis, the model ensures accurate and reliable extraction, demonstrating strong potential for real-world legal applications through consistent and reproducible results.

Additional comments

The BiLSTM-enhanced legal text extraction model using fuzzy logic and metaphor recognition showcases a cutting-edge approach to addressing complex legal language. Its integration of advanced methodologies ensures precision, adaptability, and contextual understanding, making it a valuable contribution to legal text processing and a promising tool for automating intricate legal analyses.